# *RealEngine*: SIMULATING AUTONOMOUS DRIVING IN REALISTIC CONTEXT

## ABSTRACT

Driving simulation plays a crucial role in developing reliable driving agents by providing controlled, evaluative environments. To enable meaningful assessments, a high-quality driving simulator must satisfy several key requirements: multi-modal sensing capabilities (e.g., camera and LiDAR) with realistic scene rendering to minimize observational discrepancies; closed-loop evaluation to support free-form trajectory behaviors; highly diverse traffic scenarios for thorough evaluation; multi-agent cooperation to capture interaction dynamics; and high computational efficiency to ensure affordability and scalability. However, existing simulators and benchmarks fail to comprehensively meet these fundamental criteria. To bridge this gap, this paper introduces *RealEngine*, a novel driving simulation framework that holistically integrates 3D scene reconstruction and novel view synthesis techniques to achieve realistic and flexible closed-loop simulation in the driving context. By leveraging real-world multi-modal sensor data, *RealEngine* reconstructs background scenes and foreground traffic participants separately, allowing for highly diverse and realistic traffic scenarios through flexible scene composition. This synergistic fusion of scene reconstruction and view synthesis enables photorealistic rendering across multiple sensor modalities, ensuring both perceptual fidelity and geometric accuracy. Building upon this environment, *RealEngine* supports three essential driving simulation categories: non-reactive simulation, safety testing, and multi-agent interaction, collectively forming a reliable and comprehensive benchmark for evaluating the real-world performance of driving agents.

## 1 INTRODUCTION

Autonomous driving (AD) methods (Yin et al., 2021; Li et al., 2022b; Ye et al., 2023; Hu et al., 2023b; Wen et al., 2023; Fu et al., 2024) have advanced rapidly, largely due to the introduction of diverse driving datasets (H. Caesar, 2021; Caesar et al., 2020; Montali et al., 2024) and simulators (Dosovitskiy et al., 2017). These resources facilitate research by supporting model training, testing, and evaluation across a variety of virtual and real driving environments, utilizing a wealth of multimodal sensor, including cameras and LiDAR.

However, existing AD datasets and simulators have fundamental limitations that prevent them from fully capturing real-world driving scenarios and challenges, diminishing their credibility and usefulness in practical applications. For instance, real driving data typically provides only pre-existing driving trajectories (H. Caesar, 2021; Caesar et al., 2020), allowing for open-loop evaluation without immediate feedback or interaction with the trajectories planned by driving agents. This results in a significant discrepancy between simulated and actual driving behavior. The static nature of recorded datasets, where other vehicles do not react to the actions of the ego vehicle, creates a scenario fundamentally different from real-world situations. Although the closed-loop simulator CARLA (Dosovitskiy et al., 2017) addresses this issue by providing real-time feedback to driving agents, it relies on manual 3D modeling and graphics engines, which lacks realism and exhibits a substantial appearance gap from actual driving scenarios. Consequently, models trained in this environment may struggle to handle real-world driving conditions effectively. Further, previous benchmarks primarily focus on non-collision and normal driving scenarios, which limits the models' ability to address unseen risks encountered in the real world.

Table 1: Comparison of various datasets, generative models, world models, and simulators in terms of interactivity, fidelity, diversity, and efficiency. **DATA.** represents dataset, **GEN.** represents generative model, **W.M.** represents world model, **SIM.** represents simulator.

| Type | Name | Interactivity | | | Fidelity | | Diversity | | | Efficiency |
|---|---|---|---|---|---|---|---|---|---|---|
| | | Uncontrollable closed-loop | Controllable closed-loop | Multi-agent simulation | Realistic images | Real-world roadgraph | Safety test cases | Multi-view images | LiDAR point cloud | Efficient rendering |
| DATA. | CitySim (Robinson et al., 2009) | ✗ | ✗ | ✗ | ✗ | ✗ | ✗ | ✓ | ✓ | ✓ |
| | Bench2Drive (Jia et al., 2024) | ✗ | ✗ | ✗ | ✗ | ✗ | ✗ | ✓ | ✓ | ✓ |
| | nuPlan (H. Caesar, 2021) / Navsim (Dauner et al., 2024) | ✗ | ✗ | ✗ | ✓ | ✓ | ✗ | ✓ | ✓ | ✓ |
| | nuScenes (Caesar et al., 2020) / Waymo dataset (Sun et al., 2020) | ✗ | ✗ | ✗ | ✓ | ✓ | ✗ | ✓ | ✓ | ✗ |
| GEN. | MagicDrive (Gao et al., 2023) / DriveDreamer (Wang et al., 2023a) | ✗ | ✗ | ✗ | ✓ | ✓ | ✗ | ✓ | ✗ | ✗ |
| | SimGen (Zhou et al., 2024d) | ✗ | ✗ | ✗ | ✓ | ✓ | ✗ | ✗ | ✗ | ✗ |
| W.M. | KiGRAS (Zhao et al., 2024) / SMART (Wu et al., 2024b) | ✓ | ✗ | ✓ | ✗ | ✓ | ✗ | ✗ | ✗ | ✗ |
| | MUVO (Bogdoll et al., 2023) | ✓ | ✗ | ✗ | ✗ | ✓ | ✗ | ✗ | ✗ | ✗ |
| | Vista (Gao et al., 2024) / GAIA-1 (Hu et al., 2023a) | ✓ | ✗ | ✗ | ✓ | ✗ | ✗ | ✗ | ✗ | ✗ |
| SIM. | Waymax (Gulino et al., 2024) | ✓ | ✓ | ✓ | ✗ | ✓ | ✗ | ✗ | ✗ | ✗ |
| | SUMO (Krajzewicz et al., 2012) / LimSim (Wenl et al., 2023) | ✓ | ✓ | ✗ | ✗ | ✓ | ✗ | ✗ | ✗ | ✗ |
| | CARLA (Dosovitskiy et al., 2017) | ✓ | ✓ | ✗ | ✗ | ✓ | ✗ | ✓ | ✓ | ✗ |
| | STRIVE (Rempe et al., 2022) | ✓ | ✓ | ✗ | ✗ | ✓ | ✓ | ✗ | ✗ | ✗ |
| | MetaDrive (Li et al., 2022a) | ✓ | ✓ | ✗ | ✗ | ✓ | ✓ | ✓ | ✓ | ✓ |
| | Unisim (Yang et al., 2023b) / OAsim (Yan et al., 2024a) | ✓ | ✓ | ✗ | ✓ | ✓ | ✗ | ✓ | ✓ | ✗ |
| | NeuroNCAP (Ljungbergh et al., 2024) | ✓ | ✓ | ✗ | ✓ | ✓ | ✓ | ✗ | ✗ | ✗ |
| | DriveArena (Yang et al., 2024) | ✓ | ✓ | ✗ | ✓ | ✓ | ✓ | ✓ | ✗ | ✗ |
| | HUGSIM (Zhou et al., 2024a) | ✓ | ✓ | ✗ | ✓ | ✓ | ✓ | ✓ | ✗ | ✓ |
| | *RealEngine* (Ours) | ✓ | ✓ | ✓ | ✓ | ✓ | ✓ | ✓ | ✓ | ✓ |

To advance the field, we introduce *RealEngine*, a pioneering autonomous driving simulation platform capable of rendering realistic multimodal sensor data efficiently and supporting closed-loop simulation. It is distinguished by the following features: **(i) Realistic scene rendering**, closely resembling the real world to minimize domain discrepancies between simulated and actual driving environments, allowing both camera images and LiDAR point clouds; **(ii) Closed-loop simulation**, enabling driving along free-form trajectories planned by agents while providing corresponding feedback; **(iii) Supporting diverse scenarios**, including a wide range of hazardous driving situations to facilitate comprehensive safety-critical evaluations of driving agents; **(iv) Multi-agent co-operation and interaction**, closely approximating various real-world conditions in terms of driving dynamics and scene complexity. As summarized in Table 1, no existing benchmarks or simulators meet these fundamental requirements simultaneously, as well as a spectrum of driving simulation focused features.

Our *RealEngine* is founded on the innovative concept of seamlessly integrating scene reconstruction with the composition of traffic participants. We reconstruct the background scene and foreground traffic participants separately with the corresponding real-world data. This approach simultaneously addresses key limitations of existing scene reconstruction methods (Chen et al., 2023; Wu et al., 2023; Yang et al., 2023a; Yan et al., 2024b; Chen et al., 2024; Turki et al., 2023): **(i)** Unable to acquire occluded regions within a scene; **(ii)** Inferior performance in synthesizing novel viewpoints of traffic participants; **(iii)** Limited choice of possible traffic participants. Our decomposed design naturally supports flexible scene editing while enabling the concurrent operation of multiple driving agents, as required for realistic driving simulations. By merging the background scene with diverse traffic participants, we can efficiently simulate a wide range of high-quality, unique driving scenarios tailored to any specific requirements. We generate a diverse range of driving scenarios and simulate three driving categories: *non-reactive, safety test, and multi-agent interaction*. This offers a high quality, reliable, flexible, and comprehensive benchmark for assessing the real-world performance of driving agents.

## 2 RELATED WORK

**Autonomous driving.** Recent research in autonomous driving have shifted from addressing individual tasks to exploring end-to-end planning, enabling the progressive execution from perception to ego-planning within a unified framework. Early works (Casas et al., 2021; Hu et al., 2022; Chitta et al., 2023) implement this by leveraging simplified auxiliary tasks, which limits the final performance. In contrast, UniAD (Hu et al., 2023b) and VAD (Jiang et al., 2023) have advanced this paradigm by integrating a broader spectrum of driving tasks, achieving notable progress in planning tasks by producing explicit intermediate results. Additionally, recent approaches present sparse representations (Zhang et al., 2024a; Sun et al., 2024), employ lightweight diffusion-based generation (Liao et al., 2025) to efficiently facilitate end-to-end planning. For enabling these driving methods to be evaluated extensively and authentically, we provide a reliable testing platform here.

**Sensor simulation.** The geometric reconstruction of extensive urban spaces, such as streets and cities, has played a pivotal role in autonomous driving (Sun et al., 2020; Caesar et al., 2020). Recently,

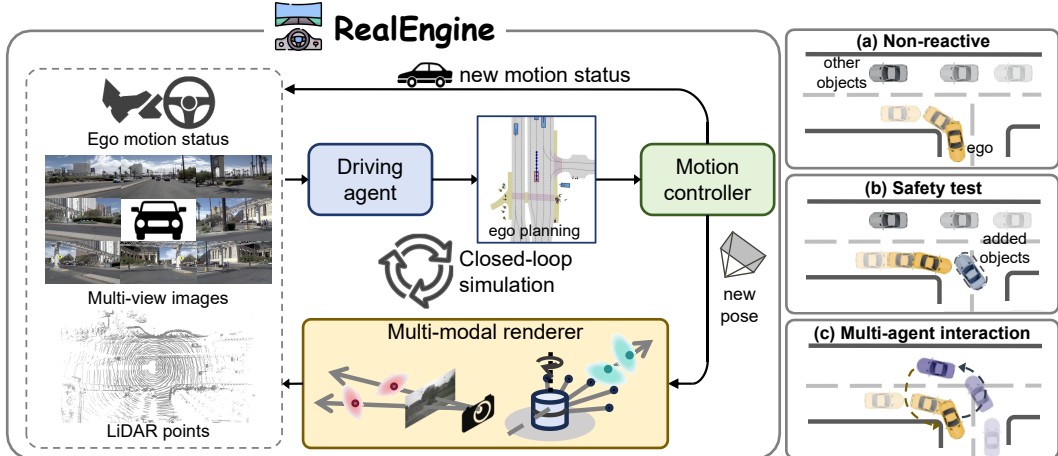

Figure 1: **The working mechanism of *RealEngine*.** It consists of three modules: a driving agent, a motion controller and a multi-model renderer. Given a traffic scene represented by multimodal data including multi-view images and LiDAR point cloud, the driving agent predicts the trajectory, according to which *RealEngine* updates the ego-motion state for all traffic participants. Moving to the next time step, the multimodal sensor data will be refreshed by the current ego-motion state, which is then used for the driving agent to make the next trajectory planing. We consider three driving situations: non-reactive, safety test, and multi-agent interaction.

Gaussian splatting (Kerbl et al., 2023) based methods have been introduced to model dynamic urban scenes. PVG (Chen et al., 2023) incorporates periodic vibration at each Gaussian primitive to represent static and dynamic objects uniformly. At the same time, explicitly decomposing scenes into independent entities has become common practice, as seen in works such as StreetGaussians (Yan et al., 2024b), DrivingGaussians (Zhou et al., 2024c), HUGS (Zhou et al., 2024b) and OmniRe (Chen et al., 2024). Recently, LiDAR simulation studies have focused on using real data for improved realism. For instance, LiDARsim (Manivasagam et al., 2020) and PCGen (Li et al., 2023) use multi-step, data-driven methods to simulate point clouds from real data. Additionally, works such as (Tao et al., 2023; Zhang et al., 2024b; Zheng et al., 2024; Xue et al., 2024; Tao et al., 2024; Wu et al., 2024a) leverage NeRF (Mildenhall et al., 2020) for scene reconstruction and LiDAR simulation. Recent works (Jiang et al., 2025; Zhou et al., 2025) introduce Gaussian splatting (Kerbl et al., 2023) into the LiDAR reconstruction task, achieving improved reconstruction quality and rendering speed. However, sensor-acquired data is prone to occlusions, leading to information loss within scenes. Additionally, reconstructed dynamic objects can distort when viewed from alternative perspectives, limiting scene editability and novel view synthesis. To address these challenges, we propose modeling the foreground and background separately allowing them to be composed in a flexible manner, resulting in a comprehensive driving simulation platform.

**Closed-loop simulation in realistic settings.** Closed-loop simulation (Gulino et al., 2024; H. Caesar, 2021; Dosovitskiy et al., 2017; Li et al., 2022a) is crucial for the evaluation and deployment of AD planning systems, collecting sufficient driving statistics and improving the emergency response capability of driving agents. To make simulators more realistic, recent research has explored using existing real-world driving datasets. Bench2Drive (Jia et al., 2024) improves the CARLA benchmark by reconstructing the data format to be more aligned with the nuScenes dataset, bridging the gap between reinforcement learning planners and end-to-end planners. Navsim (Dauner et al., 2024) imitates closed-loop evaluation to adjust the nuPlan benchmark, introducing more comprehensive and practical metrics. With the emergence of novel 3D rendering and generation research, some works integrate these techniques into existing datasets, better simulating and constructing richer and more diverse scenarios. NeuroNCAP (Ljungbergh et al., 2024) utilizes NeRF (Mildenhall et al., 2020) to render novel surrounding views and creates collision scenarios to enhance measurement. Relying on powerful generative models and extensive driving data, DriveArena (Yang et al., 2024) is capable to yield controllable and abundant real-world scenarios and realize closed-loop simulation. However, these simulators all fail to meet the fundamental criteria as mentioned earlier. This motivates the introduction of our *RealEngine*, a more reliable and powerful benchmark for assessing the real-world performance of driving agents.

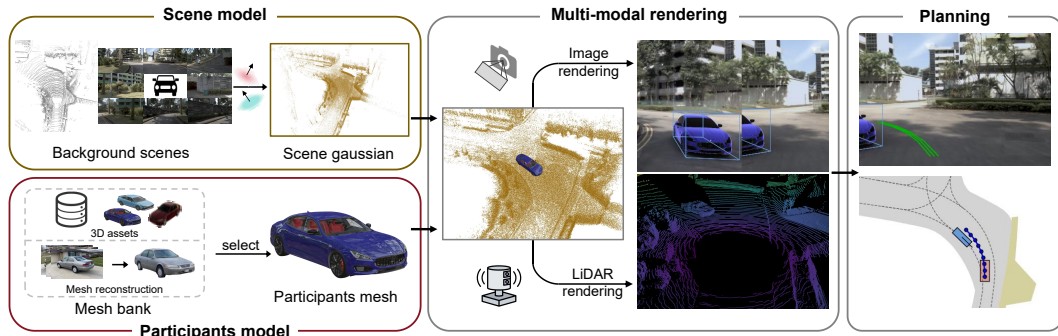

Figure 2: **Scene composition.** We start with modeling the background scene based on real sensor data and obtaining the meshes of traffic participants either by extracting them from the reconstructed data or through manual creation. That makes a rich space for designing a variety of customizable traffic scenarios. To create a specific scenario, we select both the background scene and traffic participants, which can be integrated based on each participant's spatial coordinates over time. This naturally enables the creation of highly diverse scenes to support extensive closed-loop simulation.

# 3 METHOD

We present *RealEngine*, a reliable and comprehensive autonomous driving simulation platform capable of rendering realistic multimodal sensor data efficiently and supporting closed-loop simulation. It is composed of: (i) Simulator infrastructure including background scene and traffic participants as well as their compositions; (ii) Closed-loop driving simulation; (iii) Assessment of driving agents. The working mechanism of *RealEngine* is depicted in Figure 1.

## 3.1 SIMULATOR INFRASTRUCTURE

The advancement of recent generative models has been recently exploited for closed-loop driving simulation. For example, DriveArena (Yang et al., 2024) streamlines a couple of generative functions, such as interactive traffic flows and novel scene synthesis, allowing to simulate virtual driving scenes. However, this approach suffers from several key drawbacks: (1) Need for large manually labelled training data for video generative model optimization; (2) Substantial running overhead, even without support to multimodal sensor data such as LiDAR point clouds (3) Inability to support multi-agent co-operation and interaction. (4) Dependence on a pre-defined high-definite map; (5) Low controllability on the traffic scenario including background scene and traffic participants, consequently causing frequent situational inconsistency over space and time; (6) Low spatiotemporal realism. While some of the above issues, such as realistic scene rendering, more flexible traffic scenario editing, and no need for training data collection, can be addressed by NeuroNCAP (Ljungbergh et al., 2024), it narrowly focuses on safety test of driving agents. Further, it cannot well generalize to free form trajectories otherwise the novel view synthesis will degraded substantially.

To address all the problems mentioned above, we propose decoupling the background scene and foreground traffic participants by reconstructing each of them individually from the corresponding real sensor data, allowing the composition of diverse traffic scenarios and free form driving trajectories of multiple agents while maintaining high-fidelity novel view synthesis in multiple modalities (*e.g.*, camera images and LiDAR).

### 3.1.1 BACKGROUND SCENE MODELS

To reconstruct realistic background environments, we adopt StreetGaussians (Yan et al., 2024b) for camera images and GS-LiDAR (Jiang et al., 2025) for LiDAR point clouds, chosen for their high rendering efficiency and cross-modal fidelity. Importantly, RealEngine maintains flexibility—alternative methods such as PVG (Chen et al., 2023), OmniRe (Chen et al., 2024), and LiDAR-RT (Zhou et al., 2025) can be seamlessly incorporated into our system as modular components.

Pose calibrations in nuPlan (H. Caesar, 2021) is relatively coarse, posing challenges for accurate scene reconstruction. A widely used conventional method for camera pose correction (Yan et al., 2024b; Chen et al., 2024; Hess et al., 2024) involves learning a trainable correction matrix that adjusts

the camera pose automatically during training. However, this method is only effective for small pose deviations and fails to converge on nuPlan, leading to suboptimal results. To address this issue, we propose a pose correction method based on LiDAR point cloud registration.

As the geometric information of the background scene remains fixed in the world coordinate system, we can align the ego vehicle's poses across different frames by registering LiDAR points transformed to the world. We remove dynamic vehicles from the LiDAR data based on annotation information and filter out ground points which vary significantly across frames. All remaining point clouds are transformed into the world coordinate system and truncated within a predefined region to ensure consistency across frames. We select the central LiDAR frame $\boldsymbol{P}_{\mathrm{refer}}$ as the reference frame and apply a learnable correction matrix $\boldsymbol{M}$ to transform the LiDAR frames $\boldsymbol{P}$. The transformed LiDAR frames are then compared with the reference frame using the Chamfer Distance (C-D) (Fan et al., 2017) as the loss function:

$$\mathcal{L}_{\mathrm{cd}} = \mathrm{CD}(\boldsymbol{M}\boldsymbol{P}, \boldsymbol{P}_{\mathrm{refer}}) \tag{1}$$

Additionally, we leverage learnable image exposure transformations to handle cross-camera appearance variations, and utilizes video generative prior (Yang et al., 2025; Yu et al., 2024a) to optimize the scene model across multiple trajectories in Section A.1.2. For further details on camera image and LiDAR point cloud reconstruction, please refer to Section A.1. By leveraging these advanced reconstruction techniques, we achieve precise multi-modal background reconstruction with high rendering efficiency, enabling flexible scene editing and realistic simulation of autonomous driving agents.

### 3.1.2 TRAFFIC PARTICIPANTS MODELS

Traffic participants (e.g., vehicles, bicycles) are essential for realistic driving scenarios. To enable high-fidelity and diverse simulations, we curate a comprehensive set of 3D participant models. Conventional approaches (Yan et al., 2024b; Chen et al., 2024; Zhou et al., 2024a) insert reconstructed Gaussian splatting primitives into the background scene according to predefined trajectories. While providing high-quality rendering from training viewpoints, they degrade significantly when rendering resolution, viewpoint, or object distance changes (Yan et al., 2024c; Yu et al., 2024b; Song et al., 2024). This degradation causes visual artifacts, especially in close-range interactions (e.g., collisions), and also introduces inconsistencies in lighting and shadows between foreground objects and the background scene.

To address these issues, RealEngine represents traffic participants using 3D meshes, ensuring consistent geometry across all viewpoints and distances. To achieve seamless integration, we employ a diffusion model to guide learning of scene-consistent lighting and shading, as further detailed in Section 3.1.3. Meshes are rendered using ray tracing to produce both RGB images and depth maps, enabling accurate occlusion reasoning between participants and scenes.

The 3D meshes are sourced from two complementary pipelines: (i) High-quality meshes manually created for key traffic objects. (ii) Meshes reconstructed from real-world datasets, including 360-degree images from CO3D (Reizenstein et al., 2021) and 3DRealCar (Du et al., 2024). These are processed via 3D Gaussian Splatting (3DGS) for reconstruction, followed by mesh extraction for flexible use in scene composition.

### 3.1.3 DRIVING SCENARIO COMPOSITION

The combination of reconstructed background scenes and traffic participant models forms a rich design space for creating diverse and flexible driving scenarios. Scenario composition begins by selecting a background scene, into which a set of traffic participants is inserted. Each participant is assigned an initial position, orientation, and free-form trajectory, either manually specified or dynamically planned by a driving agent. These participants are then spatially registered into the background scene's coordinate system. An overview is shown in Figure 2.

To ensure realistic integration between foreground participants and background scenes, we introduce a physically based rendering (PBR) process that optimizes environment light maps for consistent relighting and shadow casting. We adopt the Disney BRDF lighting model (Wang et al., 2023b),

where the foreground color is computed as:

$$C_{fg}(\boldsymbol{x}) = \int_{\boldsymbol{\Omega}} f_r(\boldsymbol{x}, \boldsymbol{\omega}, \boldsymbol{\omega}_o) \boldsymbol{L}_i(\boldsymbol{\omega}) |\langle \boldsymbol{\omega}, \boldsymbol{n} \rangle| \mathrm{d}\boldsymbol{\omega} \tag{2}$$

Here, $\boldsymbol{x}$ denotes the surface point where the camera ray $\omega_o$ intersects, $\boldsymbol{n}$ the surface normal, $\boldsymbol{\omega}$ the light direction, and $\boldsymbol{L}_i$ the incoming illumination. The BRDF $f_r$ models the surface reflectance.

For shadows, we assume a ground plane closely aligned to the foreground object's base. For each camera ray $\boldsymbol{\omega}_o$, we compute the ground intersection $\boldsymbol{x}'$ and its normal $\boldsymbol{n}'$. Shadow rays $\boldsymbol{\omega}'$ are traced across the hemisphere $\boldsymbol{\Omega}'$ to determine occlusion, and the shadow intensity $\boldsymbol{I}$ is computed:

$$I(\boldsymbol{x}') = \frac{\int_{\boldsymbol{\Omega}(\boldsymbol{x}')} \boldsymbol{L}_s(\boldsymbol{\omega}') |\langle \boldsymbol{\omega}', \boldsymbol{n}' \rangle| \mathrm{d}\boldsymbol{\omega}'}{\int_{\boldsymbol{\Omega}'} \boldsymbol{L}_s(\boldsymbol{\omega}') |\langle \boldsymbol{\omega}', \boldsymbol{n}' \rangle| \mathrm{d}\boldsymbol{\omega}'} \tag{3}$$

where $\boldsymbol{L}_s$ is the shadow environment map, and $\boldsymbol{\Omega}(\boldsymbol{x}')$ represents the solid angle of unoccluded light rays at point $\boldsymbol{x}'$ with respect to the foreground mesh.

The final composed image is:

$$C = C_{bg} \cdot \boldsymbol{I} \cdot (1 - \boldsymbol{M}_{fg}) + \boldsymbol{C}_{fg} \cdot \boldsymbol{M}_{fg} \tag{4}$$

where $\boldsymbol{C}_{bg}$ is the background image, $\boldsymbol{C}_{fg}$ and $\boldsymbol{M}_{fg}$ are the rendered foreground RGB and mask, respectively. We optimize $\{\boldsymbol{L}_i, \boldsymbol{L}_s\}$ using StableDiffusion (Rombach et al., 2022) with SDS loss (Poole et al., 2022), ensuring photorealistic blending.

## 3.2 CLOSED-LOOP DRIVING SIMULATION

Built upon this flexible scenario composition and novel view synthesis-enabled reconstruction, RealEngine supports a wide range of driving scenarios. We demonstrate three categories:

**Non-reactive Simulation.** Evaluates closed-loop planning where other participants follow fixed pre-recorded trajectories (Figure 1 (a)). The agent perceives rendered multi-modal sensor data, plans trajectories, and executes motion via a linear quadratic regulator, completing the closed-loop cycle.

**Safety Test Simulation.** Evaluates agent reactions to inserted participants exhibiting hazardous behaviors (*e.g.*, sudden lane changes or blocked intersections), testing safety-critical skills (Figure 1 (b)).

**Multi-agent Interaction Simulation.** Simulates cooperative and adversarial interactions where multiple agents, each running independent planning, simultaneously navigate the scene (Figure 1 (c)).

## 3.3 ASSESSMENT OF DRIVING AGENTS

Driving agent assessment aims to evaluate the agent's trajectory planning performance across both space and time, considering safety, progress, and interaction quality within diverse traffic scenarios. In traditional open-loop evaluation, the Predictive Driver Model Score (PDMS) (Dauner et al., 2024) measures trajectory quality at individual time steps. However, open-loop PDMS does not capture how planning decisions evolve over time, nor does it account for interactions with dynamic participants in the scene.

To address these issues, RealEngine introduces a *closed-loop extension* of PDMS, which evaluates the agent's performance over a continuous sequence of future steps. Specifically, starting at time step $t$, the agent plans a sequence of future positions over $N$ steps, continuously interacting with the environment and surrounding participants.

At each step, the agent receives updated sensor observations rendered from the new position, which are used to generate the next planned trajectory, creating a fully closed-loop simulation. During this process, other traffic participants follow either their pre-recorded reference trajectories or react through independent planning processes, depending on the scenario type.

The final closed-loop PDMS over the horizon $t$ to $t + N$ is formulated as:

$$\text{PDMS}_{t:t+N} = \underbrace{\left( \prod_{m \in \{\text{NC,DAC}\}} \text{score}_m \right)}_{\text{penalty terms}} \times \underbrace{\left( \frac{\sum_{w \in \{\text{EP,TTC,C}\}} \text{weight}_w \cdot \text{score}_w}{\sum_{w \in \{\text{EP,TTC,C}\}} \text{weight}_w} \right)}_{\text{weighted average rewards}}. \tag{5}$$

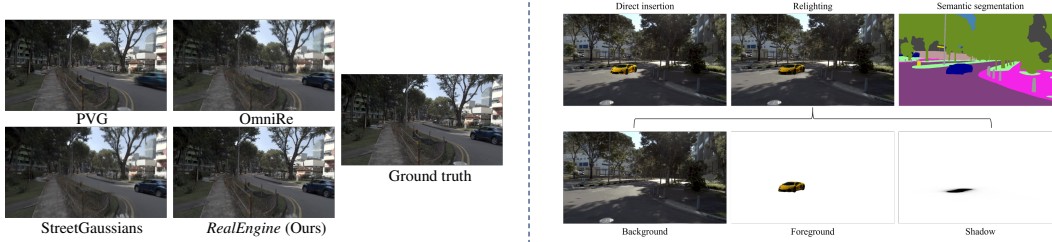

(a) Comparison of camera images reconstruction  (b) Relighting

Figure 3: (a) Compared to state-of-the-art reconstruction methods, we achieve superior camera image reconstruction in the nuPlan (H. Caesar, 2021) benchmark. Additionally, (b) our foreground relighting technique enables seamless integration of inserted participants with the background scene.

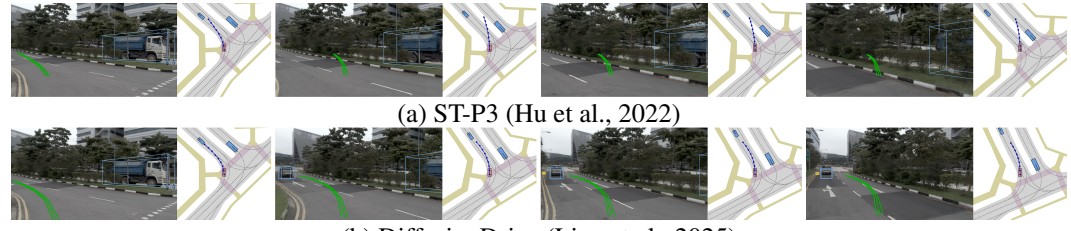

(a) ST-P3 (Hu et al., 2022)

(b) DiffusionDrive (Liao et al., 2025)

Figure 4: **Non-reactive simulation.** The driving agent's planned trajectory at each step is visualized in bird's-eye view (blue) and the front-view camera (green). During closed-loop simulation, ST-P3 (Hu et al., 2022) exhibits inconsistencies in consecutive frame planning, leading to error accumulation and causing the vehicle to navigate into invalid regions. In contrast, DiffusionDrive (Liao et al., 2025) maintains more consistent planning across consecutive frames, resulting in a higher PDM Score.

The *penalty terms* include: (1) NC (No Collision): Whether the agent avoids collisions with other vehicles, pedestrians, or cyclists. (2) DAC (Drivable Area Compliance): Whether the agent stays within the valid drivable area (lanes, intersections, etc.).

The *weighted average reward terms* reflect the overall driving quality across three dimensions: (1) EP (Ego Progress): How effectively the agent advances toward the goal within the given time horizon. (2) TTC (Time to Collision): How much time the agent maintains between itself and potential collision risks. (3) C (Comfort): Evaluating smoothness of the planned trajectory, including acceleration and jerk. This combined metric captures both safety-critical behaviors (penalties) and desirable driving qualities (rewards), providing a comprehensive assessment of the agent's closed-loop performance in diverse and dynamic traffic scenarios.

## 4 EXPERIMENTS

**Datasets.** We use CO3D (Reizenstein et al., 2021) and 3DRealCar (Du et al., 2024) to reconstruct diverse foreground traffic participants, followed by mesh extraction for rendering and scene composition. Additionally, we include high-quality meshes from Sketchfab (Contributors) for further diversity in foreground insertion and relighting.

For background scene reconstruction and scene editing, we leverage Navsim (Dauner et al., 2024), which is derived from OpenScene (Contributors, 2023), a simplified version of nuPlan (H. Caesar, 2021). We select 14 diverse sequences from Navsim for scene editing and driving agent evaluation, and design 14, 21, and 14 test cases for the non-reactive, safety, and multi-agent interaction simulation, respectively. To ensure high-quality sensor simulation, we retrieve high-frequency images and LiDAR point clouds from the corresponding nuPlan sequences for background scene reconstruction. Using an NVIDIA RTX A6000, the rendering frame rate reaches 30Hz for camera images and 15Hz for LiDAR data.

**Autonomous driving models.** We evaluate four end-to-end driving models: ST-P3 (Hu et al., 2022), VAD (Jiang et al., 2023), TransFuser (Chitta et al., 2023), and DiffusionDrive (Liao et al., 2025), with

Table 2: **Non-reactive simulation.** We show the no at-fault collision (NC), drivable area compliance (DAC), time-to-collision (TTC), comfort (Comf.), and ego progress (EP) subscores, and the PDM Score (PDMS), as percentages.

| Method | Loop | Ego stat. | Image | LiDAR | NC ↑ | DAC ↑ | TTC ↑ | Comf. ↑ | EP ↑ | PDMS ↑ |
|---|---|---|---|---|---|---|---|---|---|---|
| Constant velocity (Dauner et al., 2024) | | ✔ | | | 92.9 | 64.3 | 85.7 | 100 | 29.4 | 46.8 |
| ST-P3 (Hu et al., 2022) | Open-loop | ✔ | ✔ | | 92.9 | 71.4 | 92.9 | 100 | 46.2 | 59.6 |
| VAD (Jiang et al., 2023) | | ✔ | ✔ | | 92.9 | 85.7 | 92.9 | 100 | 48.5 | 66.1 |
| TransFuser (Chitta et al., 2023) | | ✔ | ✔ | ✔ | 92.9 | 85.7 | 92.9 | 100 | 55.9 | 69.1 |
| DiffusionDrive (Liao et al., 2025) | | ✔ | ✔ | ✔ | 92.9 | 85.7 | 92.9 | 100 | 56.7 | **69.5** |
| ST-P3 (Hu et al., 2022) | Closed-loop | ✔ | ✔ | | 100 | 64.3 | 85.7 | 100 | 35.6 | 47.5 |
| VAD (Jiang et al., 2023) | | ✔ | ✔ | | 85.7 | 78.6 | 78.6 | 100 | 34.3 | 53.0 |
| TransFuser (Chitta et al., 2023) | | ✔ | ✔ | ✔ | 92.9 | 71.4 | 85.7 | 100 | 46.0 | 57.9 |
| DiffusionDrive (Liao et al., 2025) | | ✔ | ✔ | ✔ | 92.9 | 71.4 | 92.9 | 100 | 47.1 | **61.3** |
| *Human* | | | | | *100* | *100* | *92.9* | *100* | *68.3* | *83.8* |

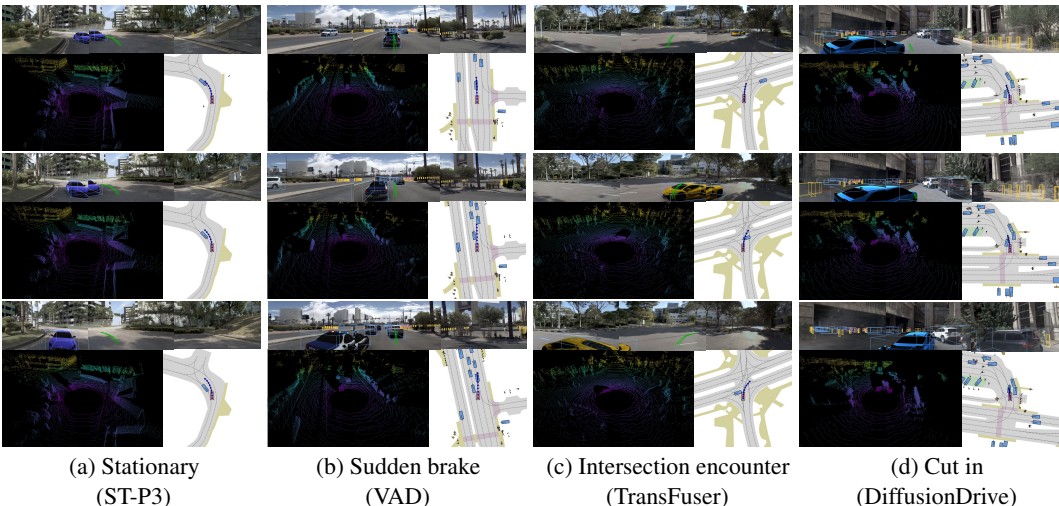

(a) Stationary (ST-P3)  (b) Sudden brake (VAD)  (c) Intersection encounter (TransFuser)  (d) Cut in (DiffusionDrive)

Figure 5: **Safety test simulation.** The driving agent's planned trajectory at each step is visualized in bird's-eye view (blue) and the front-view camera (green). The driving agent is navigating in our designed safety-critical scenarios. The agent may (a)(b) successfully avoid a collision, (c) exhibit no reaction, or (d) attempt to decelerate to prevent a collision but ultimately fail.

Table 3: **Safety test and multi-agent simulation.** The notations are consistent with the **non-reactive simulation** Table 2 above.

| Method | Simulation | Ego stat. | Image | LiDAR | NC ↑ | DAC ↑ | TTC ↑ | Comf. ↑ | EP ↑ | PDMS ↑ |
|---|---|---|---|---|---|---|---|---|---|---|
| Constant velocity (Dauner et al., 2024) | Safety test | ✔ | | | 47.6 | 71.4 | 38.1 | 100 | 36.7 | 36.3 |
| ST-P3 (Hu et al., 2022) | | ✔ | ✔ | | 47.6 | 100 | 42.9 | 100 | 44.7 | 44.4 |
| VAD (Jiang et al., 2023) | | ✔ | ✔ | | 47.6 | 95.2 | 28.6 | 100 | 41.2 | 37.0 |
| TransFuser (Chitta et al., 2023) | | ✔ | ✔ | ✔ | 47.6 | 100 | 38.1 | 100 | 44.1 | 42.2 |
| DiffusionDrive (Liao et al., 2025) | | ✔ | ✔ | ✔ | 57.1 | 100 | 52.4 | 100 | 54.0 | **53.8** |
| Constant velocity (Dauner et al., 2024) | Multi-agent | ✔ | | | 42.8 | 60.7 | 39.3 | 100 | 27.8 | 27.4 |
| ST-P3 (Hu et al., 2022) | | ✔ | ✔ | | 53.6 | 96.4 | 50.0 | 100 | 44.6 | 46.3 |
| VAD (Jiang et al., 2023) | | ✔ | ✔ | | 32.1 | 71.4 | 32.1 | 100 | 27.7 | 28.8 |
| TransFuser (Chitta et al., 2023) | | ✔ | ✔ | ✔ | 60.7 | 96.4 | 53.6 | 100 | 54.3 | **55.0** |
| DiffusionDrive (Liao et al., 2025) | | ✔ | ✔ | ✔ | 57.1 | 96.4 | 50.0 | 100 | 51.7 | 51.9 |

ST-P3 and VAD reimplemented on nuPlan (H. Caesar, 2021) for consistency. As a baseline, we also test Navsim's constant velocity model for comparison.

## 4.1 DIVING SCENARIO QUALITY

We compared our optimized reconstruction results (Section A.1) with the state-of-the-art methods (Yan et al., 2024b; Chen et al., 2024; 2023) on 6 Navsim (Dauner et al., 2024) sequences, as shown in Figure 3 (a), and observed a notable improvement in the reconstruction performance on nuPlan. Meanwhile, our foreground insertion blends more harmoniously with the background and can be recognized by perception models (Borse et al., 2021; Xie et al., 2021; Ren et al., 2024), as shown in Figure 3 (b). This enables the driving agents to correctly recognize the inserted foreground participants and respond accordingly.

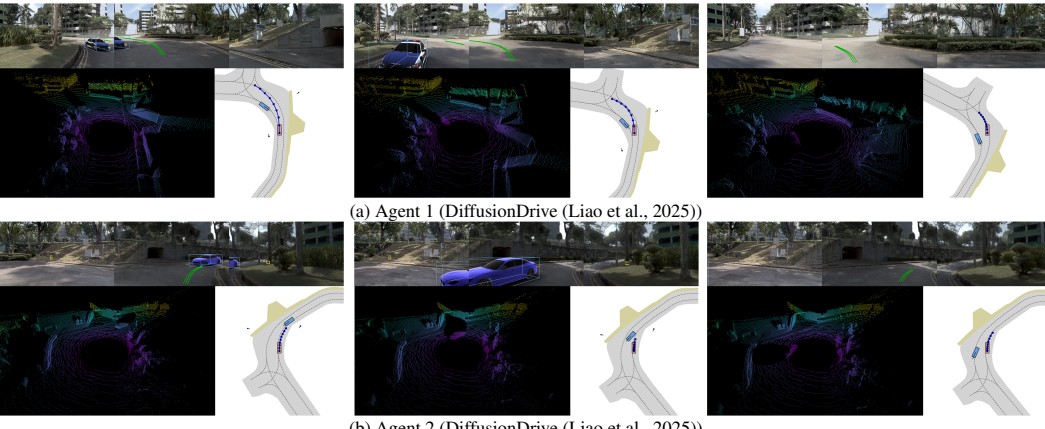

(a) Agent 1 (DiffusionDrive (Liao et al., 2025))

(b) Agent 2 (DiffusionDrive (Liao et al., 2025))

Figure 6: **Multi-agent interaction simulation.** The trajectory annotations are consistent with Figure 5. We set two agents (DiffusionDrive (Liao et al., 2025)) to plan trajectories simultaneously and independently in the showed turning scenario. The first agent avoids collision by increasing its turning radius, while the second agent decelerates to allow safe passage. Once the vehicles have passed each other, the second agent resumes its original speed.

## 4.2 CLOSED-LOOP SIMULATION

**Non-reactive simulation.** In the non-reactive setting, we conduct open-loop and closed-loop simulations on the driving agents and compare their performance, as shown in Table 2. Compared to directly predicting multiple frame trajectories in open-loop simulation, the PDM Score of the continuously predicted trajectories by the driving agent in closed-loop simulation is reduced to varying degrees. In particular, models tend to exhibit inconsistency in the adjacent predicted trajectories. This inconsistency leads to cumulative errors, causing the vehicle to navigate to unreasonable areas and resulting in a decrease in the DAC (drivable area compliance) metric, as shown in Figure 4.

**Safety test simulation.** For safety test simulation, we designed three test cases (simple, moderate, and challenging) tailored to each specific scenario and conducted closed-loop simulations to compute the PDM Score. As shown in Figure 5, in simple and moderate scenarios, such as a stationary vehicle blocking the lane or a sudden brake, driving agents can reasonably maneuver to avoid collisions. However, in more challenging scenarios, such as aggressive lane changes or intersection encounters, driving agents may exhibit some reaction but fail to avoid collisions, or in some cases, show minimal response to an imminent collision, leading to a significant drop in the PDM Score, as in Table 3.

**Multi-agent interaction simulation.** In the multi-agent interaction simulation, each vehicle is assigned to an instance of a certain driving agent model for independent and simultaneous control, with only the initial speed and high-level driving commands (right, left, straight, or unknown) specified. For each specific scenario, we design both simple and challenging test cases. At each time step, we render the sensor data for all driving agents to plan trajectories and conduct closed-loop simulations to compute the PDM scores for all instances and average the results, as presented in Table 3. Additionally, Figure 6 illustrates a turning scenario involving two DiffusionDrive (Liao et al., 2025) instances, where both agents successfully avoid collisions by adopting different yet reasonable strategies.

## 5 CONCLUSION

This paper introduces a novel driving simulation platform, *RealEngine*, capable of rendering realistic multimodal sensor data efficiently and perform closed-loop simulation in highly diverse traffic scenarios. *RealEngine* employs a foreground-background separate reconstruction and composition rendering approach, enabling convenient and more flexible control and editing of scenes. This allows for realistic interaction and continuous feedback between the driving agent and the simulation platform, while also facilitating multi-agent interactions within the simulation. Based on this, we simulate three driving categories: non-reactive, safety test, and multi-agent interaction, to establish a reliable and comprehensive benchmark for evaluating the real-world performance of driving agents.

## 6 ETHICS STATEMENT

This work focuses on reconstructing urban scenes and evaluating the performance of autonomous driving models. Our study does not involve human subjects, personally identifiable information, or sensitive private data. All datasets employed are publicly available and widely used in the research community, and we follow the corresponding licenses and usage guidelines. The reconstructed scenes and evaluation results are intended solely for academic research and do not raise immediate safety risks, as the experiments are performed in a simulation environment without deployment in real-world driving systems.

## 7 REPRODUCIBILITY STATEMENT

We are committed to ensuring the reproducibility of our work. To this end, we will release the full implementation and relevant scripts upon the final acceptance of the paper. This timeline allows us to carefully clean and document the code for ease of use. Experimental settings, hyperparameters, and implementation details are described in Section 4 and Section A.1 to facilitate verification prior to code release.

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

# A APPENDIX

## A.1 URBAN SCENE RECONSTRUCTION

### A.1.1 NUPLAN POSE CORRECTION

We propose a pose correction method based on LiDAR point cloud registration in Section 3.1.1. As shown in Figure 7, point cloud registration effectively corrects pose calibration errors. This leads to improved scene reconstruction quality, which is reflected in the increased PSNR, as presented in Table 4.

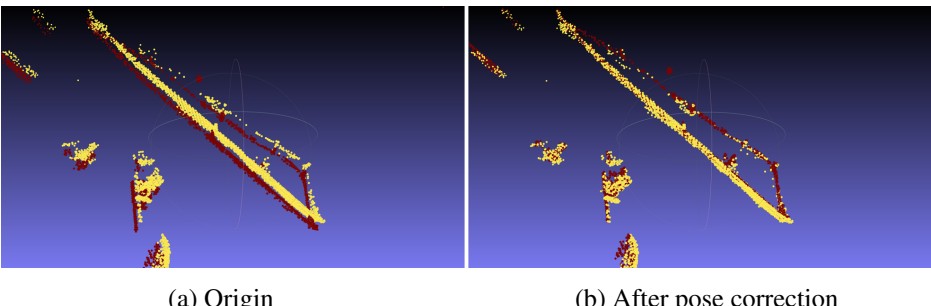

(a) Origin                (b) After pose correction

Figure 7: **Pose correction in nuPlan.** The yellow and red points originate from LiDAR point clouds of different frames. (a) Due to the coarse pose calibration in nuPlan, LiDAR point clouds from different frames are misaligned in the world coordinate system, posing challenges for high-quality reconstruction. (b) After our pose correction, LiDAR point clouds are properly aligned, leading to improved camera images reconstruction quality.

Table 4: Ablation study on camera images reconstruction in the nuPlan benchmark.

|  | PSNR↑ | SSIM↑ | LPIPS↓ |
|---|---|---|---|
| w/o $\mathcal{L}_{\mathrm{cd}}$ | 26.02 | 0.798 | 0.179 |
| w/o undistortion | 25.68 | 0.809 | 0.182 |
| w/o color correction | 28.05 | 0.853 | 0.132 |
| ***RealEngine*(Ours)** | **29.67** | **0.897** | **0.093** |

### A.1.2 CAMERA IMAGES RECONSTRUCTION

We use StreetGaussians (Yan et al., 2024b) to reconstruct the scene and perform additional processing on the nuPlan dataset. StreetGaussians (Yan et al., 2024b) uses manual pose annotation of dynamic objects to distinguish between static background and moving objects. Dynamic objects are reconstructed in their respective centered canonical spaces and then placed into the background scene space during the rendering process based on the known poses.

As shown in Figure 8 (a), the camera images in nuPlan exhibit significant barrel distortion. Since 3DGS relies on image-based rendering and does not render rays corresponding to each pixel as NeRF does, such distortion adversely affects the reconstruction of scene geometry. To mitigate this issue, we applied a distortion correction to the images. Furthermore, due to varying exposure levels across the cameras in the nuPlan dataset, the same object exhibits substantial color differences when viewed from different cameras, as shown in Figure 8 (b), leading to ambiguities in the color information of Gaussian primitives. To address this, we introduced a learnable affine transformation $\{A_i, t_i\}$ for each camera $i$ to individually calibrate the image colors:

$$\tilde{C}_i = A_i C_i + t_i \tag{6}$$

where $i$ represents the camera index, $C_i$ denotes the original rendered RGB map, and $\tilde{C}_i$ refers to the RGB map after exposure transformation. These processing methods improve the quality of scene reconstruction, as shown in the ablation study in Table 4.

We compared our optimized reconstruction results with the state-of-the-art methods (Yan et al., 2024b; Chen et al., 2024; 2023) on 6 Navsim (Dauner et al., 2024) sequences, as shown in Figure 3 and Table 5, and observed a notable improvement in the reconstruction performance on nuPlan.

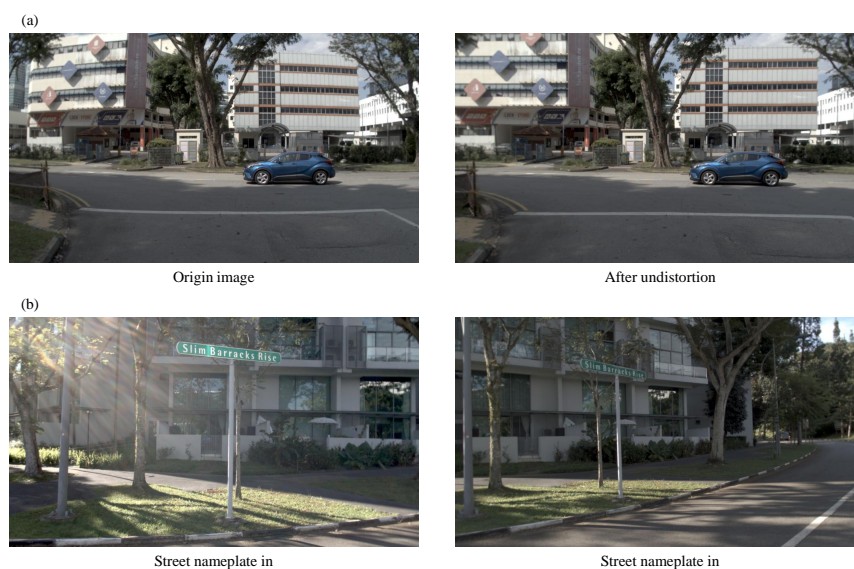

Figure 8: **Distortion and color inconsistency in the nuPlan benchmark.** (a) The camera images in nuPlan exhibit significant barrel distortion, which poses challenges for Gaussian splatting based reconstruction. To address this, we apply distortion correction to the images. (b) Additionally, different cameras have varying exposure levels. To prevent color ambiguity for the same Gaussian primitive across different cameras, we learn an exposure transformation for each camera separately.

Table 5: Comparison with state-of-the-art camera images reconstruction methods in nuPlan.

|  | PSNR↑ | SSIM↑ | LPIPS↓ |
|---|---|---|---|
| StreetGaussians (Yan et al., 2024b) | 26.02 | 0.798 | 0.179 |
| Omnire (Chen et al., 2024) | 26.89 | 0.837 | 0.165 |
| PVG (Chen et al., 2023) | 28.32 | 0.854 | 0.176 |
| *RealEngine*(**Ours**) | **29.67** | **0.897** | **0.093** |

Although existing urban reconstruction methods perform excellently in synthesizing novel viewpoints for recorded trajectories, they face challenges when handling new trajectories in closed-loop simulations due to the limited viewpoints of driving videos and the vastness of the driving environment. To address this challenge, we employ DriveX (Yang et al., 2025) to enhance the reconstructed scene, which utilizes video generative prior (Yu et al., 2024a) to optimize the scene model across multiple trajectories. As shown in Figure 9, our final scene model is capable of generating high-fidelity virtual driving environments beyond the recorded trajectories, enabling free-form trajectory driving simulations. We additionally conduct the evaluation protocol described in DriveX (Yang et al., 2025) to measure lateral offset on the nuPlan dataset. The results reported in Table 6 also shows that our method is capable of synthesizing high-quality sensor data even when the ego-vehicle deviates from the original trajectory.

Table 6: **Lane change reconstruction.** We conduct the evaluation protocol described in DriveX (Yang et al., 2025) to measure lateral offset on the nuPlan dataset.

|  | ±0m (recorded) | | ±1m | ±2m | ±3m |
|---|---|---|---|---|---|
|  | PSNR↑ | SSIM↑ | FID↓ | FID↓ | FID↓ |
| PVG (Chen et al., 2023) | 28.32 | 0.854 | 62.71 | 95.34 | 130.49 |
| StreetGaussian (Yan et al., 2024b) | 26.02 | 0.798 | 59.48 | 87.74 | 103.30 |
| DriveX (our use) (Yang et al., 2025) | **29.67** | **0.897** | **52.59** | **77.27** | **86.84** |

### A.1.3 LiDAR reconstruction

We use GS-LiDAR (Jiang et al., 2025) to reconstruct the LiDAR point clouds. GS-LiDAR (Jiang et al., 2025) introduces a novel panoramic rendering technique with explicit ray-splat intersection,

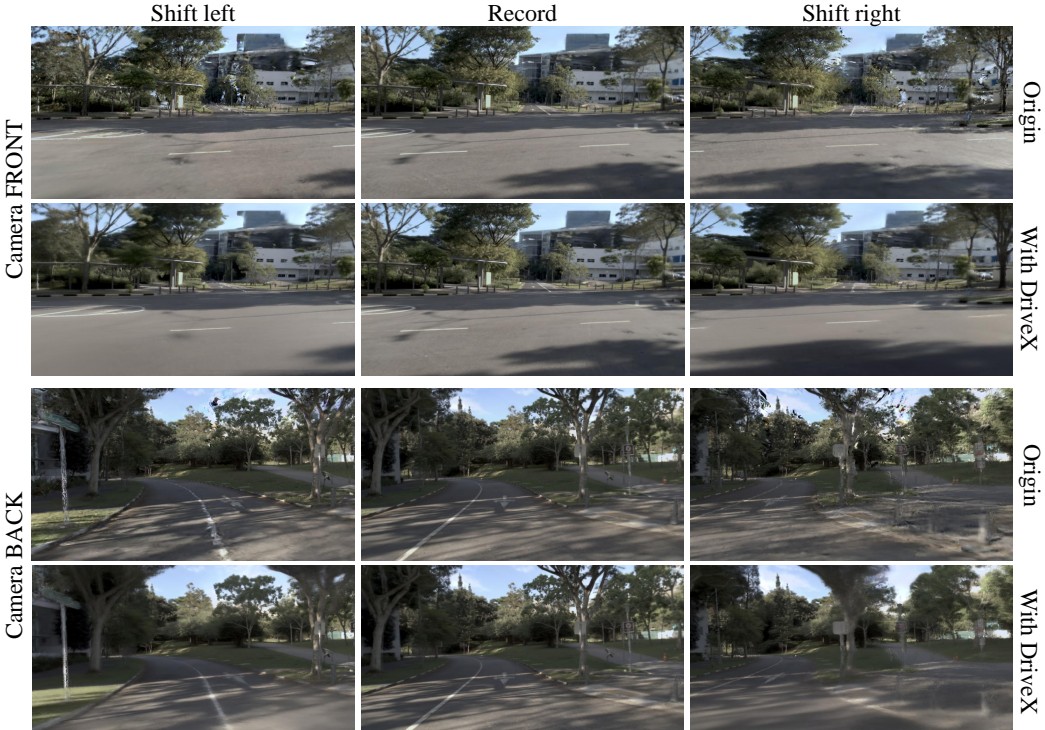

Figure 9: **Lane change reconstruction.** We render the camera images by shifting the driving perspective 3 meters to the left and right. Due to the limited viewpoint of driving videos and the vastness of the driving environment, large translations from driving perspectives lead to a significant decline in reconstruction quality. To address this, we use DriveX (Yang et al., 2025), which leverages video generative prior to optimize the scene Gaussian primitives, resulting in improved reconstruction quality even with large viewpoint translations.

guided by panoramic LiDAR supervision. This method employs 2D Gaussian primitives (Huang et al., 2024) with periodic vibration characteristics (Chen et al., 2023), allowing for precise geometric reconstruction of both static and dynamic elements in driving scenes.

The LiDAR point clouds in nuPlan (H. Caesar, 2021) is generated by merging point clouds from five LiDAR sensors, with non-uniform angular distributions across different scan lines. However, GS-LiDAR assumes that the LiDAR point cloud originate from a single laser scanner with uniformly spaced scan lines. To address this discrepancy, we reproject the merged point cloud onto the top LiDAR sensor to obtain the range map, selecting the point with the smaller depth when multiple points project to the same pixel. Additionally, we discard the non-uniform scan lines at both ends while retaining the central region, where the scan lines are evenly distributed.

As shown in Figure 10, this processing method results in some loss of LiDAR information. However, since the driving agent (Chitta et al., 2023; Liao et al., 2025) converts the LiDAR point cloud into a 2-bin histogram over a 2D BEV grid with relatively low resolution, the missing information has a minimal impact on the driving agent. We evaluate the reconstruction performance on the same 6 sequences as the camera image reconstruction in Table 7.

Table 7: Evaluation of LiDAR reconstruction performance in the nuPlan benchmark.

| | Point Cloud | Depth | | Intensity | |
|---|---|---|---|---|---|
| | CD↓ | RMSE↓ | PSNR↑ | RMSE↓ | PSNR↑ |
| GS-LiDAR (Jiang et al., 2025) | 0.315 | 6.930 | 21.25 | 0.0903 | 20.88 |

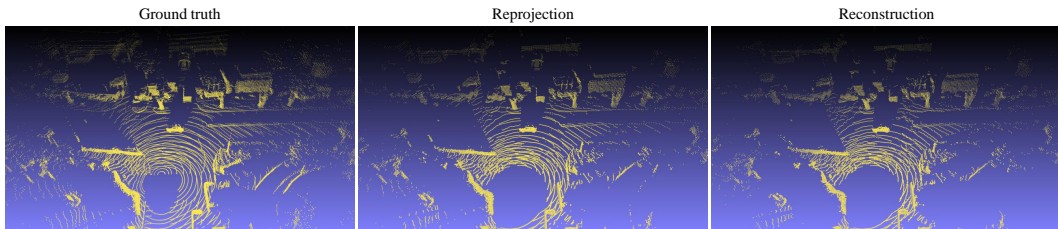

| Ground truth | Reprojection | Reconstruction |

Figure 10: **LiDAR reconstruction.** Although reprojection may lead to some loss of LiDAR information, its impact on the low-resolution histogram used by the agent is minimal. Meanwhile, GS-LiDAR (Jiang et al., 2025) achieves high-quality reconstruction of the reprojected LiDAR data.

## A.2 CONSISTENCY OF DRIVING BEHAVIORS

We replicate the same real-world evaluation scenarios from the Navsim (Dauner et al., 2024) benchmark in the non-reactive open-loop setting in Table 2, and evaluate autonomous driving models (Chitta et al., 2023; Liao et al., 2025) trained solely on real data in both the real open-loop environment and its simulated counterpart in RealEngine, comparing the performance differences.

We use the the Predictive Driver Model Score (PDMS) (Dauner et al., 2024) between trajectories as a measure of similarity, and conduct evaluations only on sequences where the PDMS is greater than zero. The gap is defined as follows:

$$\text{gap} = \frac{|\text{PDMS}_{\text{real}} - \text{PDMS}_{\text{sim}}|}{\max(\text{PDMS}_{\text{real}}, \text{PDMS}_{\text{sim}})} \tag{7}$$

Specifically, $\text{PDMS}_{\text{real}}$ refers to the PDMS of the trajectory planned based on real sensor inputs, whereas $\text{PDMS}_{\text{sim}}$ corresponds to that planned using sensor data simulated by RealEngine. To evaluate the realism of camera and LiDAR simulation independently, we test three settings: **(i)** simulating only the camera ($\text{PDMS}_{\text{cam}}$), **(ii)** simulating only the LiDAR ($\text{PDMS}_{\text{lidar}}$), and **(iii)** simulating both camera and LiDAR ($\text{PDMS}_{\text{both}}$), as shown in Table 8. To better understand the realism of RealEngine, we also visualize the trajectories planned by DiffusionDrive (Liao et al., 2025), as shown in Figure 11.

Table 8: **Consistency of driving behaviors.** We test Transfuser (Chitta et al., 2023) and Diffusion-Drive (Liao et al., 2025), which take camera and LiDAR inputs, under the three settings described in Section A.2.

| Method | Image | LiDAR | $\text{PDMS}_{\text{real}}$ ↑ | $\text{PDMS}_{\text{cam}}$ ↑ | $\text{gap}_{\text{cam}}$ ↓ | $\text{PDMS}_{\text{lidar}}$ ↑ | $\text{gap}_{\text{lidar}}$ ↓ | $\text{PDMS}_{\text{both}}$ ↑ | $\text{gap}_{\text{both}}$ ↓ |
|---|---|---|---|---|---|---|---|---|---|
| TransFuser (Chitta et al., 2023) | ✔ | ✔ | 81.18 | 80.99 | 0.77% | 80.78 | 0.55% | 80.56 | 0.98% |
| DiffusionDrive (Liao et al., 2025) | ✔ | ✔ | 80.75 | 81.07 | 0.89% | 80.85 | 0.42% | 81.07 | 1.07% |

The results show that the planned trajectory gap between the real-world environment and the counterpart simulated by RealEngine remains small (approximately 1%), which highlights the realism of our simulator and its effectiveness in supporting closed-loop evaluation of autonomous driving models.

Moreover, to provide a quantitative downstream evaluation, we test the performance of the perception module of DiffusionDrive (Liao et al., 2025) with respect to the ground-truth foreground inserted. Specifically, we measuring the IoU between the model predicted bounding boxes and the ground-truth annotations on the original dataset, as well as the IoU between the model predicted bounding boxes and the inserted foreground annotations for the inserted objects. A smaller gap between the two IoUs indicates that the inserted foreground objects are more realistic and better perceived by the autonomous driving model.

As shown in Table 9, although our relighting is not yet perfect, the inserted foreground objects can still be effectively perceived by the autonomous driving model. Moreover, the model's ability to react to and avoid the inserted objects in certain simple scenarios further supports the realism of our insertions.

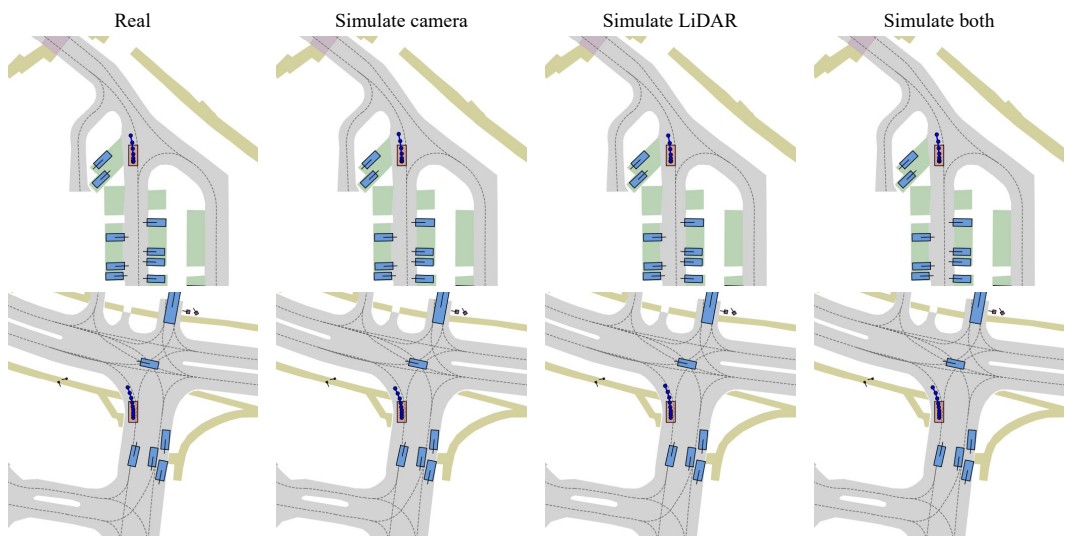

Figure 11: **Consistency of driving behaviors.** We test DiffusionDrive (Liao et al., 2025) under the same settings in Table 8. The close alignment between trajectories in the real and simulated environments demonstrates the high fidelity of our RealEngine.

Table 9: **Consistency of perception module**. We evaluate the Intersection over Union (IoU) between the model-predicted bounding boxes of DiffusionDrive (Liao et al., 2025) and the ground-truth annotations in the original dataset, as well as the IoU between the model predictions and the annotations of the inserted foreground objects. A smaller discrepancy between these two IoU values suggests that the inserted objects are more realistic and are better recognized by the autonomous driving model.

| | IoU-origin↑ | IoU-inserted↑ | gap↓ |
|---|---|---|---|
| DiffusionDrive (Liao et al., 2025) | 72.3 | 68.9 | 4.7% |

### A.3 THE USE OF LARGE LANGUAGE MODELS

In preparing this manuscript, we employed a large language model (LLM) solely as a writing assistance tool. Specifically, the LLM was used to polish and refine the grammar, style, and clarity of English sentences drafted by the authors. All research ideas, experimental design, analyses, and substantive writing were entirely conducted by the authors. The LLM did not contribute to research ideation, methodology, results, or conclusions.

### A.4 ADDITIONAL SIMULATION RESULTS

We provide more detailed simulation videos in the supplementary materials. Please refer to **simulation.mp4** for the non-reactive, safety test, and multi-agent interaction simulation.

