# OpenReview forum: "RealEngine: Simulating Autonomous Driving in Realistic Context"
_ICLR.cc/2026/Conference — Submitted to ICLR 2026_

### Official Review · Reviewer_sAct · 2025-10-17

**Soundness:** 2
**Presentation:** 1
**Contribution:** 1
**Rating:** 2
**Confidence:** 5

**Summary:**

This paper proposes a simulation framework for closed-loop benchmarking of autonomous driving agents. It leverages real-world driving data to reconstruct the background and combine it with 3D meshes of foreground traffic participants to composite a realistic and reactive driving scenario. It shows the application of such a framework in non-reactive simulation, safety testing, and multi-agent interaction.

**Strengths:**

1. Closed-loop simulation is important for benchmarking end-to-end driving models, and simulating using real-world data helps address the sim2real gap.
2. The composition of the reconstructed static background and foreground objects with 3D meshes is flexible and could help with closed-loop simulation.

**Weaknesses:**

1. The paper is hard to follow, with the contribution not clearly stated. Especially in the method section, why specifically compare to DriveArena and mention its drawbacks? The proposed method is not comparable to DriveArena with fundamental differences. The proposed method is based on reconstructing real-world data, while DriveArena uses a generative model to synthesize observations. The main difference from the previous approaches should be the separate construction of foreground objects and the background from my understanding.
2. The main component of the simulation framework is the scene reconstruction and composition module, but it mainly relies on existing approaches without much technical novelty.
3. The space and time horizon of the simulation shown in the supplementary video is very limited, and I doubt whether such a short duration is really useful for closed-loop testing, especially for the multi-agent interaction scenario.
4. From the qualitative video, it seems the framework only simulates vehicles, but other essential traffic participants, like pedestrians, are not simulated. The traffic light is also not simulated, which is common at the intersections.
5. The proposed evaluation benchmark consists of only 14 sequences from Navsim, which lacks diversity. Therefore, the results from such a benchmark may not be general and comprehensive for the driving agent's performance.

**Questions:**

1. How does the proposed method help to better acquire occluded regions within a scene as mentioned in the introduction?
2. How does the chamfer distance loss help to align the point cloud? Point clouds from different timesteps could capture different parts of the scene, and aligning them only with the chamfer distance may be suboptimal?
3. Can the proposed framework include pedestrian simulation as well?
4. Is the ego-agent used in simulation the same one for data capture? Can we put an additional ego-vehicle in the reconstructed scene?
5. Can the proposed framework be easily extended to other datasets besides NuPlan? Will the reconstruction quality be degraded given different sensor configurations?
6. Can the authors elaborate more on optimizing the shadow and lightning of the inserted assets with the SDS loss from diffusion models? The current blending of foreground objects still looks very unnatural.

---

### Official Review · Reviewer_VuT4 · 2025-10-27

**Soundness:** 3
**Presentation:** 3
**Contribution:** 3
**Rating:** 8
**Confidence:** 4

**Summary:**

This paper presents RealEngine, a Gaussian-splatting–based simulator for autonomous driving. RealEngine reconstructs static 3D scenes via 3D Gaussian splatting and pairs them with photorealistically rendered, controllable foreground traffic to generate high-fidelity scenarios. The pipeline enables end-to-end, non-iterative closed-loop evaluation on the nuPlan dataset and reports comparative results across popular end-to-end driving methods.

**Strengths:**

- `Addresses a key evaluation gap:` As end-to-end autonomous driving becomes more prevalent, rigorous evaluation remains challenging because intermediate perception outputs are not exposed. The proposed system enables non-iterative closed-loop evaluation without relying on intermediate signals, which is a substantive technical contribution.

- `Cohesive and well-engineered pipeline:` The overall system design appears sound, and the implementation effort is substantial and appreciated.

- `Benchmarking value:` The experimental results on the simulator are informative and position the framework as a promising benchmark for standardized comparison of end-to-end driving methods.

**Weaknesses:**

- `Foreground–background domain gap and realism (Fig. 2):`
The participant model in Fig. 2 appears visually inconsistent with the reconstructed background even after relighting, creating a noticeable distribution gap. Models may exploit these artifacts to “spot” foreground objects, potentially inflating performance relative to real images. I wish to see the idea behind this concern from the authors.

- `Novel-trajectory rendering and temporal consistency:`
It’s unclear how well 3DGS rendering holds when the ego vehicle follows trajectories not present in nuPlan logs. Novel-view extrapolation can suffer from floaters, holes, or “texture swimming.”
Please include visualizations along genuinely novel paths (lateral offsets, new turns).

- `Safety-critical scenario support and policy impact: `
The paper focuses on evaluation, but it is important to know whether RealEngine can author and leverage rare/hazardous events for training end-to-end policies.

**Questions:**

N/A

---

### Official Review · Reviewer_GUJq · 2025-11-01

**Soundness:** 3
**Presentation:** 2
**Contribution:** 2
**Rating:** 2
**Confidence:** 4

**Summary:**

This paper presents RealEngine, a system-level framework for realistic and interactive autonomous driving simulation.
RealEngine integrates multiple existing techniques — Gaussian-splatting for static background reconstruction, LiDAR-based pose correction, mesh-based dynamic objects, and diffusion-guided relighting — into a unified multi-modal simulation engine that supports both camera and LiDAR rendering and closed-loop evaluation.
Experiments on nuPlan sequences demonstrate higher image reconstruction quality (PSNR/SSIM/LPIPS) than previous methods such as PVG and StreetGaussians, and a small PDMS gap (~1%) between real-world and simulated trajectories, indicating strong behavioral consistency. The paper positions RealEngine as a potential benchmark for closed-loop evaluation.


While RealEngine is a promising engineering contribution with solid implementation and clear motivation, it lacks sufficient methodological novelty and theoretical insight expected for ICLR. The paper would be better suited for a systems-oriented venue  rather than a core machine learning conference.

**Strengths:**

●System Integration: A well-engineered pipeline that combines multiple state-of-the-art components (3DGS, GS-LiDAR, PBR rendering, diffusion relighting) into a coherent simulator.
●Practical Value: Addresses a clear gap between open-loop datasets and realistic, interactive simulation for autonomous driving.
●Quantitative Rigor: Provides detailed ablation studies, pose-correction evaluation, and multi-agent PDMS comparisons.
●High-quality Visual Results: Demonstrates strong image reconstruction metrics and visually appealing renderings.

**Weaknesses:**

●Limited Novelty: The paper mainly integrates existing techniques rather than introducing a novel algorithmic contribution. The background reconstruction uses StreetGaussians and GS-LiDAR, the foreground uses off-the-shelf meshes, and the composition uses a PBR pipeline with a diffusion prior. The proposed "innovations," such as LiDAR-based pose correction and exposure compensation, feel more like necessary engineering adjustments for the nuPlan dataset rather than fundamental research contributions. The paper reads more like a well-executed system engineering project than a research paper introducing a novel algorithmic insight.
●Insufficient Technical Depth: Key modules—such as diffusion-guided relighting and GS-LiDAR integration—are described only at a high level without clear mathematical formulation or ablation. The relighting pipeline lacks details on optimization objectives and diffusion loss design, while the multi-modal fusion is treated as a black-box step, offering little technical insight or reproducibility.
●Incomplete Evaluation:
○No scalability analysis for multi-agent or long-sequence simulation.
○Limited exploration of edge cases (e.g., nighttime, adverse weather).
○Behavioral metric (PDMS) is domain-specific and lacks comparison to broader RL/AD simulation benchmarks.
●Dependence on External Assets: The quality of foreground meshes relies heavily on existing datasets (CO3D, 3DRealCar). This dependence limits scalability and generalization, as the paper provides no details on asset adaptation or automated mesh generation.

**Questions:**

see the weakness section.

---

### Official Review · Reviewer_EJhm · 2025-11-01

**Soundness:** 2
**Presentation:** 3
**Contribution:** 3
**Rating:** 4
**Confidence:** 3

**Summary:**

This paper introduces RealEngine, a comprehensive simulation framework for autonomous driving that aims to bridge the critical gap between the high flexibility of traditional simulators and the high-fidelity realism of real-world datasets. The authors identify key limitations in existing tools: datasets are non-reactive (open-loop), while simulators like CARLA suffer from a significant sim-to-real gap.

Building on this powerful foundation, the authors establish a benchmark with three essential evaluation modes: (1) non-reactive simulation with static background traffic, (2) safety testing with injected critical scenarios, and (3) multi-agent interaction where multiple planners operate simultaneously. The paper demonstrates the platform's capabilities by evaluating several state-of-the-art driving models, providing a detailed analysis of their performance in these challenging, realistic, and interactive scenarios.

**Strengths:**

Strength
1.	The paper addresses a fundamental and highly significant problem in autonomous driving research. The lack of a simulator that is simultaneously realistic, interactive, and controllable is a major bottleneck for developing and validating robust driving agents. RealEngine represents a substantial step forward in addressing this need, and its contribution is very timely given the recent advances in neural rendering.
2.	A standout strength of this work is its comprehensiveness. The authors do not just propose a renderer; they deliver a complete simulation platform and benchmark. The three proposed simulation modes—non-reactive, safety test, and multi-agent—cover the most critical use cases for evaluating modern driving planners. The extensive experiments, testing four different driving models and providing both quantitative (Tables 2 & 3) and qualitative (Figures 4, 5, 6) results, convincingly demonstrate the utility and power of the RealEngine framework.
3.	The paper provides strong evidence of the simulator's realism. The visual results in Figure 3 show a clear improvement in reconstruction quality over prior work. More importantly, the quantitative analysis in Section A.2 (Table 8), which shows a mere ~1% gap in planner performance between the real world and the simulation, is a powerful testament to the low sim-to-real gap achieved by the system.

**Weaknesses:**

Weaknesses:
1.	The paper accurately identifies a critical need for a realistic, closed-loop simulator. However, the solution can be characterized as a sophisticated systems integration effort rather than a work of fundamental algorithmic novelty. It combines SOTA rendering techniques effectively but does not propose new rendering methods or a more insightful data generation paradigm. More critically, while 'multi-agent interaction' is highlighted as a key capability, this contribution feels significantly underdeveloped. The paper provides a high-level demonstration but omits crucial details about the implementation. The most valuable contribution for the community would be the methodology for defining and controlling these interactions—for instance, the framework for modeling cooperative, adversarial, or game-theoretic behaviors between agents. Without these details, it is unclear how researchers can use RealEngine to conduct a principled study of agent-agent dynamics, which arguably is one of the most challenging frontiers in autonomous driving. This lack of detail leaves one of the paper's most exciting claims without sufficient scientific backing.
2.	The framework relies on a library of 3D meshes for traffic participants. The paper mentions these are sourced from manual creation and reconstruction from datasets like CO3D. This asset creation process is a known bottleneck. The paper would be strengthened by a more detailed discussion on the scalability of this pipeline. How sensitive is the realism and the downstream planner performance to the diversity and quality of this mesh library? A limited library could inadvertently become a source of domain bias
3.	RealEngine provides powerful control over agent trajectories, making it an excellent testbed for specific scenarios and multi-agent interactions. However, the problem of populating a scene with diverse, realistic, and reactive background traffic is not fully addressed. The "non-reactive" mode uses pre-recorded trajectories, which is a standard but limited approach. Simulating the long tail of complex human driving behaviors at scale remains an open challenge that is outside the scope of this work but is worth acknowledging as a limitation.

**Questions:**

na

---

### Meta-Review · Area_Chair_jszt · 2026-01-07

**Summary:**

The primary review comments I relied on were the many repeated concerns of novelty, absent technical details, and incomplete evaluation.

**Reviewer Concerns:**

There was no rebuttal submitted, therefore all concerns by reviewers are outstanding.

**Reviewer Scores:**

As there was no rebuttal submitted, I believe reviewers would have read each other's reviews and decided on a final recommendation of rejection. The primary reason for this is the many repeated concerns of novelty, absent technical details, and incomplete evaluation.

---

### Decision · Program_Chairs · 2026-01-26

Reject